# Towards Category Unification of 3D Single Object Tracking on Point Clouds

**Jiahao Nie[1], Zhiwei He[1✉], Xudong Lv[1], Xueyi Zhou[2], Dong-Kyu Chae[2], Fei Xie[3]**

[1]School of Electronics and Information, Hangzhou Dianzi University, China

[2]School of Computer Science, Hanyang University, Korea

[3]MoE Key Lab of Artificial Intelligence, AI Institute, Shanghai Jiao Tong University, China

`{jhnie,zwhe,lvxudong}@hdu.edu.cn, jaffe031@sjtu.edu.cn`

`{hokyeejau,dongkyu}@hanyang.ac.kr`

## Abstract

Category-specific models are provenly valuable methods in 3D single object tracking (SOT) regardless of Siamese or motion-centric paradigms. However, such over-specialized model designs incur redundant parameters, thus limiting the broader applicability of 3D SOT task. This paper first introduces unified models that can simultaneously track objects across all categories using a single network with shared model parameters. Specifically, we propose to explicitly encode distinct attributes associated to different object categories, enabling the model to adapt to cross-category data. We find that the attribute variances of point cloud objects primarily occur from the varying size and shape (*e.g.*, large and square vehicles *v.s.* small and slender humans). Based on this observation, we design a novel point set representation learning network inheriting transformer architecture, termed *AdaFormer*, which adaptively encodes the dynamically varying shape and size information from cross-category data in a unified manner. We further incorporate the size and shape prior derived from the known template targets into the model's inputs and learning objective, facilitating the learning of unified representation. Equipped with such designs, we construct two category-unified models SiamCUT and MoCUT. Extensive experiments demonstrate that SiamCUT and MoCUT exhibit strong generalization and training stability. Furthermore, our category-unified models outperform the category-specific counterparts by a significant margin (*e.g.*, on KITTI dataset, $\sim$12% and $\sim$3% performance gains on the Siamese and motion paradigms).

## 1 Introduction

Artificial general intelligence (AGI), an emerging concept in the field of artificial intelligence (AI), aims to achieve versatile cognitive abilities akin to human intelligence. AGI is envisioned as an intellectual system capable of managing a wide range of tasks. Recently, with the advancement of deep learning technology, increasing efforts (Ghiasi et al., 2021; Girdhar et al., 2022) have been devoted to exploring general vision models that can simultaneously address diverse vision tasks.

As a fundamental task in computer vision, 3D single object tracking (SOT) on LiDAR point clouds holds significant potential for various application, such as autonomous driving, mobile robotics, and augment reality (Xie et al., 2023; Lv et al., 2021; Zhang & Tao, 2020). Given a point cloud sequence, with an arbitrary object in the first frame serving as a template target, the goal of tracking is to search for this target in subsequent successive frames. Currently, 3D SOT methods can be mainly divided into two paradigms: Siamese paradigm (Giancola et al., 2019; Qi et al., 2020; Hui et al., 2021; Zhou et al., 2022; Nie et al., 2023b) and motion-centric paradigm (Zheng et al., 2022a). The former utilizes an appearance matching mechanism to determine seed points, subsequently inferring the target's position based on these points. The latter considers the predicted result from the previous frame as the target prior, and then infers the target's relative displacement between the previous and current frames. Both paradigms have achieved outstanding performance. However, regardless of the paradigm followed, existing methods commonly perform the tracking task for each object category independently, as illustrated in Fig. 1 (a). This requires the models to learn and evaluate on individual datasets specific to each category, leading to limited generalization and redundant parameters. Therefore, a question naturally arises: *Is it possible to track objects across all categories by a unified model with shared parameters?*

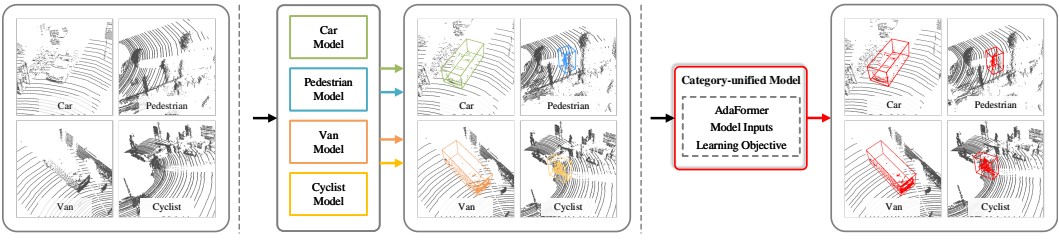

(a) Previous category-specific models  (b) Our category-unified models

Figure 1: Comparison between different tracking models. In previous category-specific models (a), multiple networks are required to perform individual tracking task for each category. In contrast, our category-unified models (b) can simultaneously track objects across all categories using a single network with shared parameters.

Through a series of comprehensive empirical studies, we discover that training models directly with cross-category data leads to suboptimal generalization and training stability, *i.e.*, tracking performance is relatively unstable after the model converges (detailed in Section 4.3). This phenomenon can be primarily attributed to three key factors incurred by the changing size and shape of cross-category objects. First, existing tracking models (Qi et al., 2020; Shan et al., 2021; Zheng et al., 2021; Hui et al., 2022; Nie et al., 2023c) employ a point set network (Qi et al., 2017a;b) as their backbone, where receptive field of each point is expanded via a group operation. However, such a group operation typically acts on a 3D sphere with a fixed radius, resulting in fixed receptive field that is not adaptable to changing size and shape information embedding; Second, search region of current frame is generated by enlarging predicted target area from previous frame by a fixed distance. This imposes an inconsistent proportion of foreground and background within the search regions for different object categories (*i.e.*, more background interference for smaller objects, less for larger objects); Third, offset learning is widely available in both the Siamese and motion-centric paradigms (Qi et al., 2020; Zheng et al., 2022a), and there is a considerable gap in the distribution of offset outputs across cross-category objects due to the different sizes and shapes. A similar problem occurs in the distribution of positive and negative samples. Consequently, different categories of data exhibit distinct learning objectives.

Based on the aforementioned observations, we introduce SiamCUT and MoCUT, which can simultaneously track objects across all categories using a single network with shared model parameters, as shown in Fig. 1 (b). Our unified models consist of three core designs: a unified representation network, model inputs and learning objective. We first design a novel point set representation network, termed *AdaFormer* that serves as the representation network for our unified tracking models. Our *AdaFormer* block consists of a deformable group vector-attention sub-block to aggregates information over a variable range of receptive fields. To be specific, this sub-block incorporates a group regression module, which learns adaptive group configurations to accommodate shape- and size-changed geometric information from cross-category data. Additionally, a vector-attention mechanism (Zhao et al., 2021) is utilized to facilitate feature interaction of points within the resulting groups. Then, in order to form unified model inputs, we leverage the known size and shape of the template target to guide the generation of search regions. These search regions are generated by cropping 3D point cloud scene at a scale that corresponds to the area occupied by the template target. As a result, the unified inputs maintain a consistent proportion of foreground and background, effectively mitigating the impact on feature learning caused by variations in background distributions across different object categories. Furthermore, we associate the predicted objects' offsets in 3D space with their length, width and height, thereby unifying model's learning objective and effectively enhancing the stability and generalization of the proposed unified models.

A comprehensive set of experiments are conducted on two challenging benchmarks, including KITTI (Geiger et al., 2012) and NuScenes (Caesar et al., 2020). The experimental results demonstrate that our category-unified models can allow for the tracking of all object categories using a single network. Moreover, it is important to note that our models not only exhibit stable generalization, but also outperform category-specific counterparts across all categories. Next, our **key contributions** can be summarized as: **1)** We propose two category-unified models SiamCUT and MoCUT based on Siamese and motion-centric 3D SOT paradigms, respectively. To the best of our knowledge, this is the first work to unify tracking across all object categories. **2)** We design a novel point set network, termed *AdaFormer*, which encodes geometric information of different object cat-

egories in a unified manner, enabling the proposed unified models to adapt to cross-category data. In addition, we further introduce unified model inputs and learning objective, which facilitate the learning of unified representation and thereby enhancing the generalization and stability of our models. **3)** To verify our unified models, extensive experiments are conducted on KITTI and NuScenes datasets. Our SiamCUT and MoCUT demonstate excellent performance with strong generalization and training stability. Furthermore, given that our category-unified models perform better than existing category-specific models, *we may enlighten the community to perform 3D SOT by cross-category training and testing instead of a per-category basis*.

## 2 RELATED WORK

### 2.1 CATEGORY-SPECIFIC TRACKING

In recent years, object tracking has been divided into two categories: multiple object tracking (MOT) and single object tracking (SOT). Depending on different data patterns, tracking models can be further categorized into those based on 2D camera images or 3D LiDAR point clouds. 2D MOT aims to simultaneously estimate identities and bounding boxes of an indeterminate number of objects with pre-defined categories. Tracking-by-detection paradigm (Zhang et al., 2022; Wu et al., 2023a; 2022; 2023b) has witnessed significant success by associating detection results from consecutive frames. Nevertheless, owing to the variations in semantic information of different object categories, the detector and association components of the model cannot be shared across categories. Thus, 2D MOT models are inherently category-specific, requiring specific datasets. For instance, MOT17 (Milan et al., 2016) and MOT20 (Dendorfer et al., 2020) are tailored to track multiple pedestrians in crowded streets. 3D MOT (Zhu et al., 2022; Ding et al., 2023) shares with 2D MOT a comparable task definition and model structure. Differently, 3D MOT is primarily geared towards tracking vehicles or pedestrians in autonomous driving road scenarios (Geiger et al., 2012; Caesar et al., 2020). 3D SOT focuses on tracking a single object in a point cloud sequence of continuous frames, where object is pre-categorized into a predefined set of categories within relevant datasets like KITTI and NuScenes. Current methods (Giancola et al., 2019; Qi et al., 2020; Zheng et al., 2021; Shan et al., 2021; Hui et al., 2021; Zhou et al., 2022; Zheng et al., 2022a; Hui et al., 2022; Nie et al., 2023c;d;b) for 3D SOT typically adopt Siamese or motion-centric paradigms to perform tracking on a per-category basis. Despite the demonstrated success, category-specific tracking results in redundant parameters. Therefore, unifying categories for tracking is a hot issue.

### 2.2 CATEGORY-UNIFIED TRACKING

Image-based 2D SOT can be classified into the scope of generic object tracking (GOT) (Zheng et al., 2022b), which is designed to track an arbitrary object given in the initial frame of a video sequence, regardless of its category (Javed et al., 2022). In contrast to MOT, which relies on the extraction of category-level semantic information for distinguishing object categories and subsequent data association, 2D SOT predominantly adopts Siamese matching paradigm (Bertinetto et al., 2016; Li et al., 2018; Zhang et al., 2020; Nie et al., 2022b;a; 2023a; Cui et al., 2022). Given the inherent advantage of Siamese representation networks that are free of learning category-specific semantic information, 2D SOT can effectively track arbitrary objects using a single model.

Inspired by the concept of GOT, we target at developing category-unified 3D SOT. Nevertheless, directly applying techniques (Chen et al., 2021; Xie et al., 2022) from 2D SOT for category unification in 3D SOT poses significant challenge. In 2D SOT, objects are shaped into a standardized size through scaling and resizing images, without impacting the color and texture information of the objects. In contrast, such normalization operations in 3D SOT seriously distort the geometric information of point clouds. Moreover, LiDAR point clouds are usually sparse, textureless, and semantically incomplete, making it difficult to encode generic features like highly-structured images.

### 2.3 DYNAMIC GROUPING

To achieve category-unified tracking, the model is required to encode varying geometric properties, such as size and shape of different object categories in a unified manner. However, existing point set networks (Qi et al., 2017a;b) utilized in 3D SOT face limitations attributed to the fixed receptive field resulting from size-fixed grouping. In the context of detection, some dynamic grouping operations (Mao et al., 2021; Yang et al., 2022; Wang et al., 2022b;a) have been proposed. *E.g.*, DBQ-SSD (Yang et al., 2022) assigns suitable receptive field for each point based on the corresponding point features. CAGroup3D (Wang et al., 2022a) introduces a class-aware local group strategy

to capture object-level shape diversity. Motivated by these approaches, our *AdaFormer* integrates a deformable group regression module to learn adaptive groups for diverse object categories.

## 3 METHODOLOGY

### 3.1 OVERVIEW

**Task Definition.** Given a category-known template target denoted as $\mathcal{P}^t = \{p_i^t\}_{i=1}^{N_t}$, along with its corresponding 3D bounding box (BBox) $\mathcal{B}_t = (x_t, y_t, z_t, w_t, h_t, l_t, \theta_t)$ in the initial frame, 3D SOT aims to locate this object within search region $\mathcal{P}^s = \{p_i^s\}_{i=1}^{N_s}$ and yield a tracking BBox $\mathcal{B}_s = (x_s, y_s, z_s, \theta_s)$ frame by frame. Here, $N_t$ and $N_s$ denote the number of points for template and search region, respectively. $(x, y, z)$ and $(w, h, l)$ represent the center coordinate and size, while $\theta$ is the rotation angle around $up$-axis. Note that, due to the consistent size of given target across all frames, only 4 parameters $(x, y, z, \theta)$ are required to be predicted for $\mathcal{B}_s$.

**Revisiting Siamese and Motion-centric Paradigms.** To predict the parameter set $(x, y, z, \theta)$, existing approaches adopt either Siamese or motion-centric paradigms to design category-specific models that are trained using data corresponding to individual category. Siamese network based models typically extract geometric features of template and search region using a Siamese backbone (Qi et al., 2017b). The extracted features are then fused to generate seed points for subsequent target localization. The localization process is executed via a 3D region proposal network (RPN), as introduced in VoteNet (Qi et al., 2019). This RPN structure serves to offset the seed points towards the target's 3D center and predicts a rotation angle and a classification score for each point, thereby yielding $M$ 3D proposals $\{(x_i, y_i, z_i, \theta_i)\}_{i=1}^{M}$ and corresponding proposal-wise scores $\{s_i\}_{i=1}^{M}$. The proposal with the highest score is considered as the tracking result. On the other hand, motion-centric models directly concatenate the point clouds $\mathcal{P}_{j-1}^t = \{p_i^t\}_{i=1}^{N_t}$ from the previous frame with the point clouds $\mathcal{P}_j^s = \{p_i^s\}_{i=1}^{N_s}$ of the current frame, and then segment the foreground points by a point set network. After that, a PointNet (Qi et al., 2017a) is employed to infer the relative movement $(\Delta x_t, \Delta y_t, \Delta z_t, \Delta \theta_t)$ of the target to be tracked. In summary, both the Siamese and motion-centric paradigms adhere a general structure characterized as "feature extraction + point offsetting", which can be formulated as:

$$(x_s, y_s, z_s, \theta_s) = \mathcal{O}_{offset}(\mathcal{F}_{siamese/motion}(\{p_i^t\}_{i=1}^{N_t}, \{p_i^s\}_{i=1}^{N_s})), \qquad (1)$$

where $\mathcal{F}$ and $\mathcal{O}$ represent feature embedding and offset embedding, respectively.

**Problem Description.** In this paper, we propose to explore category-unified model for 3D SOT. However, a significant challenge emerges when attempting to train models directly using cross-category data. Our investigation reveals that the feature learning network $\mathcal{F}_{siamese/motion}$ encounters difficulty in adapting to diverse object categories with varying size and shape attributes. Moreover, the inherent attributes variations of cross-category data are associated with distinct learning objectives, thereby posing distraction for the offset learning module $\mathcal{O}_{offset}$. Therefore, we view the category-unified tracking as two fundamental sub-problems, *i.e.*, the learning of a unified feature embedding and a unified offset embedding.

**Proposed Solution.** To tackle these challenges, we first propose a novel point set network (*AdaFormer*, Section 3.2), which effectively encodes shape- and size-changed geometric information from cross-category in a unified manner. To facilitate the learning of unified feature embedding, we further design unified model inputs (Section 3.3), which ensures that the foreground-background ratios within the input data remain invariant across different object categories. In addition, a unified learning objective (Section 3.4) is devised, including the consistent numerical distribution of predicted targets and the balanced distribution of positive and negative samples. To this end, we construct two category-unified tracking models, *i.e.*, SiamCUT and MoCUT.

### 3.2 UNIFIED REPRESENTATION NETWORK: ADAFORMER

The overall architecture of our *AdaFormer*, as illustrated in Fig. 2, is constituted by a series of cascaded subsample operators and *AdaFormer* blocks. The core design is deformable group vector-attention, which aims to extract unified feature representation for cross-category objects. More concretely, it utilizes a group regression module to learn deformable groups to enable dynamically adaptive receptive fields, and then performs feature interaction of points within these deformable groups via a vector-attention mechanism. The details are presented below.

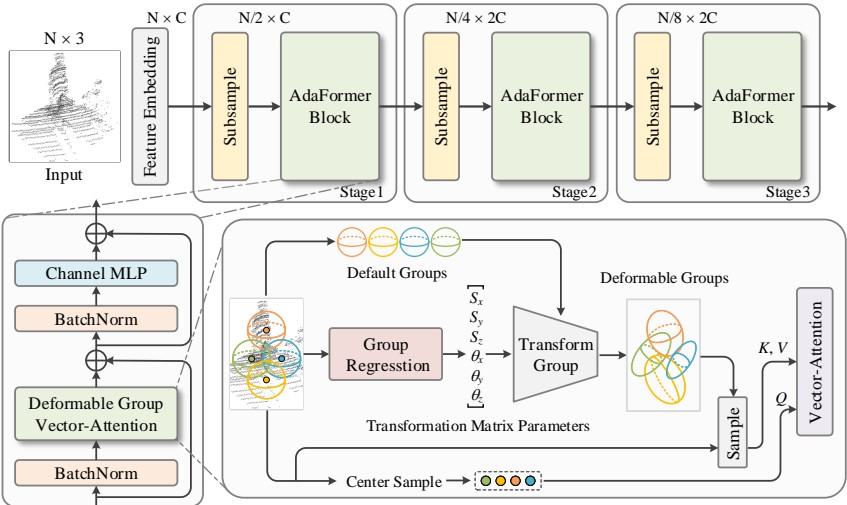

Figure 2: **Overall architecture of *AdaFormer*.** The proposed unified representation network shares a similar three-stage hierarchical structure with existing point set network (Qi et al., 2017b) used in 3D SOT, consisting of a series of subsample operators and *AdaFormer* blocks. Our representation network is empowered to learn dynamic groups through a deformable group vector-attention sub-block, thereby enabling a variable range of receptive fields.

**Deformable Group Vector-Attention.** Given input points $\{c_i = [x_i, y_i, z_i]\}_{i=1}^n$ along with their corresponding features $\{f_i\}_{i=1}^n$, we treat these points as 3D centers and generate a set of spheres with a fixed radius $r$, which serve as default groups $\{\mathcal{G}_i^r\}_{i=1}^n$. To enable dynamically adaptive receptive fields for objects across different categories, we introduce a group regression module. This module regards the default groups as references and applies a projection transformation to each group, resulting in deformable groups $\{\mathcal{G}_i^{def}\}_{i=1}^n$ with varying sizes and shapes. Specifically, the projection transformation is defined as a combination of scaling and rotation along the $xyz$ axis, involving a scaling matrix $T_s$ and three rotation matrices $T_{r_x}$, $T_{r_y}$ and $T_{r_z}$:

$$
T_s = \begin{bmatrix} s_x & 0 & 0 \\ 0 & s_y & 0 \\ 0 & 0 & s_z \end{bmatrix}, \quad T_{r_x} = \begin{bmatrix} 1 & 0 & 0 \\ 0 & \cos\theta_x & -\sin\theta_x \\ 0 & \sin\theta_x & \cos\theta_x \end{bmatrix},
$$
$$
T_{r_y} = \begin{bmatrix} \cos\theta_y & 0 & -\sin\theta_y \\ 0 & 1 & 0 \\ \sin\theta_y & 0 & \cos\theta_y \end{bmatrix}, \quad T_{r_z} = \begin{bmatrix} \cos\theta_z & -\sin\theta_z & 0 \\ \sin\theta_z & \cos\theta_z & 0 \\ 0 & 0 & 1 \end{bmatrix},
$$

(2)

where the scaling parameters $s_x$, $s_y$, $s_z$ and the rotation parameters $\theta_x$, $\theta_y$, $\theta_z$ need to be predicted to create transformation matrices. Towards this goal, our group regression module predicts 6-dimensional vectors $\{t_i = [s_{x_i}, s_{y_i}, s_{z_i}, \theta_{x_i}, \theta_{y_i}, \theta_{z_i}]\}_{i=1}^n$ for the default groups $\{\mathcal{G}_i^r\}_{i=1}^n$. E.g., the vector $t_j$ for $j$-th group $\mathcal{G}_j^r$ can be calculated as:

$$
t_j = \text{MLP}(\text{AveragePool}(\{[f_i; c_i]\}_{i=1}^k)), \quad [f_i; c_i] \in \mathcal{G}_j^r
$$

(3)

where $k$ is the number of points gathered within the $j$-th group $\mathcal{G}_j^r$ from the previous layer, and $t_j$ is learned based on prior information of the features and coordinates of the points within this group. To this end, final transformation matrix is calculated by sequentially multiplying the derived scaling matrix and rotation matrices:

$$
T = T_s \times T_{r_x} \times T_{r_y} \times T_{r_z}
$$

(4)

By using Eq. 4, a series of transformation matrices $\{T_i\}_{i=1}^n$ are generated. For the $j$-th group $\mathcal{G}_j^r$, we project the coordinates of all input points $\{c_i = [x_i, y_i, z_i]\}_{i=1}^n$ to a new 3D coordinate space using $j$-th transformation matrix $T_j$:

$$
\{[x_i^{new}, y_i^{new}, z_i^{new}]^\top\}_{i=1}^n = T_j \times \{[x_i, y_i, z_i]^\top\}_{i=1}^n
$$

(5)

Afterwards, a unit sphere is used to gather $k$ points that are closest to the center coordinate $[x_j, y_j, z_j]$ within this sphere, thereby forming a new group $\mathcal{G}_j^{def}$. By this way, we obtain a series of deformable groups $\{\mathcal{G}_i^{def}\}_{i=1}^n$, which offer dynamic receptive fields tailored to diverse object categories.

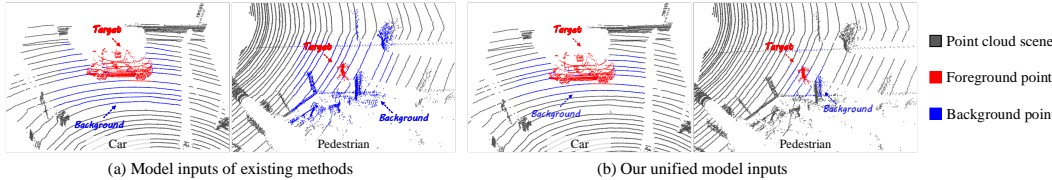

(a) Model inputs of existing methods        (b) Our unified model inputs

Figure 3: Comparison between different search regions, *i.e.*, model inputs. The previous methods (a) generate search regions by expanding the predicted result of previous frame by a fixed 3D distance, while our method (b) expands it by a scale to the width, height and length of target objects.

To form a unified feature representation, we utilize vector-attention to facilitate feature learning and interaction of points within each group $\mathcal{G}_i^{def}$. Unlike traditional self-attention, vector-attention not only preserves the permutation invariance of point clouds but also effectively models both channel and spatial information interactions. In detail, we regard the features of center point within $\mathcal{G}_j^{def}$ as query, while the features of other points as key and value. Additionally, position embedding is obtained by:

$$p_j = \omega_p(c_j - \{c_i\}_{i=1}^k), \quad c_i \in \mathcal{G}_j^{def} \tag{6}$$

where $\omega_p$ represents a two-layer MLP. Leveraging the query, key, value and position embedding, the process of feature learning and interaction for $\mathcal{G}_j^{def}$ can be mathematically expressed as:

$$f_j^{va} = \mathrm{VA}(q_j, k_j, v_j, p_j) = \mathrm{Softmax}(\varphi(\omega_q(q_j) - \omega_k(k_j) + p_j)) \cdot (\omega_v(v_j) + p_j) \tag{7}$$

where $\omega_q$, $\omega_k$, $\omega_v$ and $\varphi$ denote linear layers. The original features $f_j$ are transformed into new features $f_j^{va}$ by integrating information within the deformable receptive field.

## 3.3 UNIFIED MODEL INPUT

In the tracking procedure, to ensure that the search area of each frame contains the target object to be tracked while avoiding the inclusion of excessive irrelevant background information, a common practice in existing methods (Qi et al., 2020; Zheng et al., 2022a; Nie et al., 2023b) is to expand the area occupied by the predicted result of the previous frame by a certain distance to generate search regions. Consequently, representation network needs to simultaneously learn geometric information of objects and distinguish between foreground and background points. In fact, background information is inevitably introduced during tracking. However, such a practice leads to inconsistent foreground-background ratios of search regions, especially for different categories of objects. As shown in 3 (a), a relatively large car and a relatively small pedestrian exhibit significantly different background information, which distracts the representation network.

A similar issue occurs with single object tracking based on 2D images. To generate unified model inputs for performing generic object tracking on arbitrary objects, 2D SOT methods (Bertinetto et al., 2016; Li et al., 2018; Chen et al., 2021) typically obtain the region of interest by adding a consistent margin context $p$ centered on the target object. Subsequently, the region is resized to a constant rectangle $A^2$ using a resizing operator $s$:

$$A^2 = s(w + p) \times s(h + p) \tag{8}$$

where $w$ and $h$ denote the width and height of the target object. The amount of margin context $p$ is set to $\frac{w+h}{2}$. For point cloud tracking, however, employing similar operators will distort geometry structure of point clouds. Fortunately, benefiting from the proposed representation network *AdaFormer* is able to adapt to changing shape and size information. Therefore, instead of focusing on the amount of background context, we aim to unify the foreground-background ratios. To achieve this, we expand the area of the predicted result of the previous frame by adding a scale of width, height and length to its size:

$$H \times W \times L = (w + \alpha \cdot w) \times (h + \alpha \cdot h) \times (l + \alpha \cdot l) \tag{9}$$

where $\alpha$ denotes the scale factor, and we can form unified model inputs with consistent foreground-background ratios, as shown in Fig. 3 (b). This eliminates interference with the learning of our unified representation network.

### 3.4 UNIFIED LEARNING OBJECTIVE

In addition to achieving unified feature representation through our proposed *AdaFormer* network, a unified learning objective also matters for category-unified models. As illustrated in Eq. 1, the tracking model involves point offsets. However, different object categories possess distinct prediction targets and exhibit varied positive-negative sample distributions, which prevents a category-unified model from effectively tracking all categories of objects.

To address these problems, we propose to develop a consistent numerical distribution of predicted targets, as well as a balanced distribution of positive and negative samples. Specifically, we first correlate the specific offset values along $xyz$ axis in 3D space with the width $w$, height $h$ and length $l$ of the target object:

$$[\Delta x, \Delta y, \Delta z] = [w, h, l] \cdot [x^l, y^l, z^l] \tag{10}$$

where $w$, $h$ and $l$ are known, given by the template target in the initial frame, and the prediction targets $[\Delta x, \Delta y, \Delta z]$ are transformed into $[x^l, y^l, z^l]$. As a result, we obtain the unified numerical distribution of prediction targets, *i.e.*, offsets for objects across different categories.

Then, to balance positive and negative samples and enable a unified positive-negative sample distribution during the training phase, we introduce shape-aware labels. Different from existing methods that define points at a certain distance from the target center as positive samples and other points as negative samples, causing an imbalanced distribution of positive and negative samples for cross-categories data, our method defines the points within a cube scaled of the target object as positive samples:

$$\ell = \begin{cases} 1 & \text{if } (x_i, y_i, z_i) \in \mathbb{R}^{\beta w \times \beta h \times \beta l}, \\ 0 & \text{otherwise}, \end{cases} \tag{11}$$

where $\beta$ is the scale factor. Combining these two designs, we achieve unified learning objective, which facilitates the training of our category-unified models.

## 4 EXPERIMENT

### 4.1 EXPERIMENTAL SETTING

**Datasets.** To evaluate our category-unified models, we utilize two popular datasets KITTI (Geiger et al., 2012) and NuScenes (Caesar et al., 2020) to conduct comprehensive experiments. KITTI contains 21 training sequences and 29 test sequences. Due to the inaccessibility of the test labels, we split the training sequences into training and testing sets following previous works (Giancola et al., 2019; Zheng et al., 2022a). NuScenes is a more challenging and large-scale dataset, which contains 700 and 150 scenes for training and testing, respectively.

**Evaluation Metrics.** Following common practice in previous methods (Giancola et al., 2019; Qi et al., 2020), we measure *Success* and *Precision* metrics in One Pass Evaluetion (OPE) manner (Wu et al., 2013) to evaluate the tracker. *Success* calculates the intersection over union (IOU) between the predicted BBox and the ground truth BBox, while *Precision* calculates the distance between the centers of the two BBoxes.

**Implementation Details.** Here, we propose a unified representation network, model inputs and learning objective. To construct category-unified models, we integrate these three components into existing classic trackers P2B (Qi et al., 2020) and M$^2$Track (Zheng et al., 2022a), following Siamese and motion-centric paradigms. To be a fair comparison, we use consistent parameters with setting of P2B and M$^2$Track, including model hyper-parameters and training parameters. In addition, the scale factors $\alpha$ and $\beta$ brought by our methods are set to 1.0 and 0.4, respectively. All experiments are conducted using these parameters if not specified.

### 4.2 COMPARISON WITH STATE-OF-THE-ART METHODS.

**Results on KITTI.** Tab. 1 compares the proposed category-unified methods with other state-of-the-art category-specific methods on the KITTI dataset. Our unified models exhibit excellent tracking performance, confirming the feasibility and potential of category-unified model design. MoCUT achieves average *Success* and *Precision* of 65.8% and 85.0% by using a single model, only slightly lower than the latest CXTrack. In addition, compared to the baseline methods P2B and M$^2$Track, our SiamCUT and MoCUT not only significantly reduce the number of parameters, but also effectively track objects of different categories using a single network. Moreover, we obtain improved

performance across all categories, attributed to the proposed unified components. Notably, the performance improvements for categories with relatively small data, including Pedestrian, Van and Cyclist, are higher than Car category with relatively large data. This is due to the inherent benefit of cross-category training, where an increase training sample size will contribute to enhanced performance.

Table 1: Performance comparisons with state-of-the-art methods on KITTI dataset. **Bold** and underline represent our category-unified methods and the corresponding baselines. "      " and "      " refer to Siamese and motion-centric paradigms, respectively.

| Method | Source | Car [6,424] | Pedestrian [6,088] | Van [1,248] | Cyclist [308] | Mean [14,068] |
|---|---|---|---|---|---|---|
| SC3D (Giancola et al., 2019) | CVPR2019 | 41.3 / 57.9 | 18.2 / 37.8 | 40.4 / 47.0 | 41.5 / 70.4 | 31.2 / 48.5 |
| P2B (Qi et al., 2020) | CVPR2020 | 56.2 / 72.8 | 28.7 / 49.6 | 40.8 / 48.4 | 32.1 / 44.7 | 42.4 / 60.0 |
| MLVSNet (Wang et al., 2021) | ICCV2021 | 56.0 / 74.0 | 34.1 / 61.1 | 52.0 / 61.4 | 34.4 / 44.5 | 45.7 / 66.6 |
| PTT (Shan et al., 2021) | IROS2021 | 67.8 / 81.8 | 44.9 / 72.0 | 43.6 / 52.5 | 37.2 / 47.3 | 55.1 / 74.2 |
| V2B (Hui et al., 2021) | NeurIPS2021 | 70.5 / 81.3 | 48.3 / 73.5 | 50.1 / 58.0 | 40.8 / 49.7 | 58.4 / 75.2 |
| PTTR (Zhou et al., 2022) | CVPR2022 | 65.2 / 77.4 | 50.9 / 81.6 | 52.5 / 61.8 | 65.1 / 90.5 | 57.9 / 78.2 |
| TAT (Lan et al., 2022) | ACCV2022 | 72.2 / 83.3 | 57.4 / 84.4 | 58.9 / 69.2 | 74.2 / 93.9 | 64.7 / 82.8 |
| STNet (Hui et al., 2022) | ECCV2022 | 72.1 / 84.0 | 49.9 / 77.2 | 58.0 / 70.6 | 73.5 / 93.7 | 61.3 / 80.1 |
| DMT (Xia et al., 2023) | T-ITS2023 | 66.4 / 79.4 | 48.1 / 77.9 | 53.3 / 65.6 | 70.4 / 92.6 | 55.1 / 75.8 |
| GLT-T (Nie et al., 2023c) | AAAI2023 | 68.2 / 82.1 | 52.4 / 78.8 | 52.6 / 62.9 | 68.9 / 92.1 | 60.1 / 79.3 |
| OSP2B (Nie et al., 2023b) | IJCAI2023 | 67.5 / 82.3 | 53.6 / 85.1 | 56.3 / 66.2 | 65.6 / 90.5 | 60.5 / 82.3 |
| CXTrack (Xu et al., 2023) | CVPR2023 | 69.1 / 81.6 | 67.0 / 91.5 | 60.0 / 71.8 | 74.2 / 94.3 | 67.5 / 85.3 |
| SyncTrack (Ma et al., 2023) | ICCV2023 | 73.3 / 85.0 | 54.7 / 80.5 | 60.3 / 70.0 | 73.1 / 93.8 | 64.1 / 81.9 |
| M²Track (Zheng et al., 2022a) | CVPR2022 | 65.5 / 80.8 | 61.5 / 88.2 | 53.8 / 70.7 | 73.2 / 93.5 | 62.9 / 83.4 |
| SiamCUT | Ours | **58.1 / 73.9** | **48.2 / 76.2** | **63.1 / 74.9** | **36.7 / 47.4** | **54.0 / 74.6** |
| MoCUT | Ours | **67.6 / 80.5** | **63.3 / 90.0** | **64.5 / 78.8** | **76.7 / 94.2** | **65.8 / 85.0** |

**Results on NuScenes.** Tab. 2 presents the comparison results on the NuScenes dataset, which provides large-scale and more challenging scenes for 3D SOT. Our methods demonstrate consistent performance advantages, when the benefit of cross-category training is absent, owing to the large-scale data in most categories, which further validates the effectiveness of our proposed category-unified models. Additionally, we observe that SiamCUT shows a larger performance improvement compared to the baseline than MoCUT, in both the KITTI and NuScenes datasets. This phenomenon can be attributed to the fact that the Siamese paradigm relies on sophisticated shape and size information embedding modules for tracking, while the motion-centric paradigm directly infers relative displacement between two frames. The latter is relatively simpler and less susceptible to the changes in shape and size information when dealing with cross-category data.

Table 2: Performance comparisons with state-of-the-art methods on nuScenes dataset. **Bold** and underline represent our category-unified methods and the corresponding baselines.

| Method | Car [64,159] | Pedestrian [33,227] | Truck [13,587] | Trailer [3,352] | Bus [2,953] | Mean [117,278] |
|---|---|---|---|---|---|---|
| SC3D (Giancola et al., 2019) | 22.31 / 21.93 | 11.29 / 12.65 | 30.67 / 27.73 | 35.28 / 28.12 | 29.35 / 24.08 | 20.70 / 22.20 |
| P2B (Qi et al., 2020) | 38.81 / 43.18 | 28.39 / 52.24 | 42.95 / 41.59 | 48.96 / 40.05 | 32.95 / 27.41 | 36.48 / 45.08 |
| PTT (Shan et al., 2021) | 41.22 / 45.26 | 19.33 / 32.03 | 50.23 / 48.56 | 51.70 / 46.50 | 39.40 / 36.70 | 36.33 / 41.72 |
| PTTR (Zhou et al., 2022) | 51.89 / 58.61 | 29.90 / 45.09 | 45.30 / 44.74 | 45.87 / 38.36 | 43.14 / 37.74 | 44.50 / 52.07 |
| GLT-T (Nie et al., 2023c) | 48.52 / 54.29 | 31.74 / 56.49 | 52.74 / 51.43 | 57.60 / 52.01 | 44.55 / 40.69 | 44.42 / 54.33 |
| M²Track (Zheng et al., 2022a) | 55.85 / 65.09 | 32.10 / 60.72 | 57.36 / 59.54 | 57.61 / 58.26 | 51.39 / 51.44 | 49.32 / 62.73 |
| SiamCUT (Ours) | **40.96 / 44.91** | **31.42 / 53.80** | **53.91 / 52.65** | **63.29 / 58.21** | **41.03 / 38.01** | **40.41 / 48.54** |
| MoCUT (Ours) | **57.32 / 66.01** | **33.47 / 63.12** | **61.75 / 64.38** | **60.90 / 61.84** | **57.39 / 56.07** | **51.19 / 64.63** |

## 4.3 ABLATION STUDIES

In this section, we present a series of ablation studies, including the proposed components within our unified models and the analysis of generalization and stability. Following previous works (Qi et al., 2020; Zheng et al., 2022a), all ablated experiments are conducted on the KITTI dataset.

**Components of Unified Models.** To give a better understanding of our unified category-unified models, we investigate the impact of the designed unified components to the tracking performance. As reported in Tab. 3, removing any component will lead to a significant decrease in average performance for the Siamese paradigm. In addition, for the motion-centric paradigm, although it has a simpler structure and does not overly rely on size and shape information for tracking, our unified components still effectively enable the model to uniformly process cross-category data, thus enhancing performance. *More detailed analysis of these components can be referred to Appendix.*

**Generalization and Stability Analysis.** As illustrated in Fig. 4, the proposed unified components enable the model to learn generalizable features by explicitly encoding distinct shape and size information in a unified manner, thus guiding performance improvements across all categories. Besides,

Table 3: Ablation studies for different components of Siamese and motion-centric paradigms on KITTI dataset. The last rows represent the full category-unified models.

| Paradigm | Unified Representation Network: AdaFormer | Unified Model Inputs | Unified Learning Objective | Car [6,424] | Pedestrian [6,088] | Van [1,248] | Cyclist [308] | Mean [14,068] |
|---|---|---|---|---|---|---|---|---|
| Siamese | ✗ | ✓ | ✓ | 56.3 / 72.4 | 33.2 / 60.4 | 57.0 / 68.6 | 32.2 / 43.5 | 45.9 / 63.2 |
| | ✓ | ✗ | ✓ | 57.9 / 73.9 | 38.0 / 66.7 | 61.6 / 72.8 | 34.1 / 44.1 | 49.1 / 70.1 |
| | ✓ | ✓ | ✗ | 57.5 / 72.8 | 44.5 / 72.1 | 61.2 / 70.4 | 35.6 / 44.5 | 51.8 / 71.7 |
| | ✓ | ✓ | ✓ | **58.1 / 73.9** | **48.2 / 76.2** | **63.1 / 74.9** | **36.7 / 47.4** | **54.0 / 74.6** |
| Motion | ✗ | ✓ | ✓ | 65.2 / 78.8 | 61.9 / 88.6 | 59.5 / 74.4 | 74.1 / 92.7 | 63.5 / 83.0 |
| | ✓ | ✗ | ✓ | 67.5 / 80.4 | 63.0 / 89.3 | 63.8 / 77.9 | 76.6 / 83.4 | 65.5 / 84.2 |
| | ✓ | ✓ | ✗ | 66.9 / 80.1 | 63.2 / 88.7 | 62.4 / 77.0 | 74.3 / 93.0 | 65.1 / 83.9 |
| | ✓ | ✓ | ✓ | **67.6 / 80.5** | **63.3 / 90.0** | **64.5 / 78.8** | **76.7 / 94.2** | **65.8 / 85.0** |

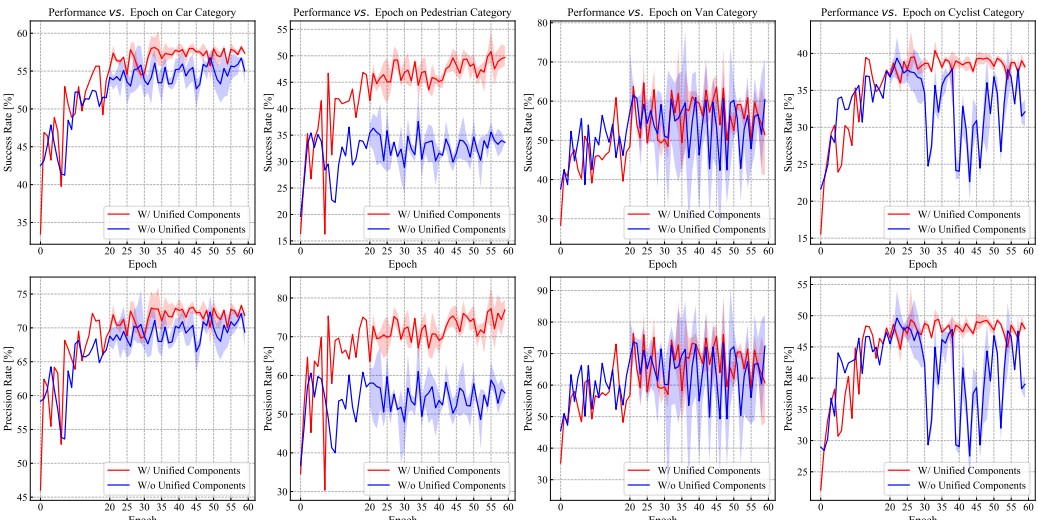

Figure 4: Generalization and stability comparisons of with and without the proposed unified components. We plot "performance *v.s.* epoch" curves on four categories, and include error bands calculated by running the corresponding experiments three times using different random seeds.

we observe that the blue curves and error bands after 20 epochs (the model tend to converge after 20 epochs), show relatively large jitter in the case of "without unified components". This instability is ascribed to the fact that the model is disturbed by cross-category data. Fortunately, our unified components alleviate this instability to some extent.

## 4.4 INFERENCE SPEED

Computational efficiency is a critical aspect in practical applications. To evaluate the tracking speed of our proposed category-unified models, we adopt a standard implementation (Qi et al., 2020; Nie et al., 2023c) and calculate the tracking speed by counting the average running time of all frames in the Car category of the KITTI dataset. SiamCUT and MoCUT run at real-time speed of 36.4 and 47.9 frame per second (Fps), respectively on a single NVIDIA 3070Ti GPU, including 7.2/8.6 ms for point cloud preprocessing, 19.8/12.0 ms for network forward computation, and 0.5/0.3 ms for post-processing.

## 5 CONCLUSION

We propose SiamCUT and MoCUT, two category-unified models to address 3D single object tracking (SOT) on point clouds task. For the first time, our methods achieve the unification of network architecture and cross-category data training and testing, allowing us to simultaneously track objects across all categories using a single network with shared parameters. Extensive experiments demonstrate that both SiamCUT and MoCUT perform better than existing category-specific counterparts on two challenging KITTI and NuScenes datasets. We hope our work will inspire further exploration of category-unified tracking architecture for the 3D SOT task.

## ACKNOWLEDGEMENTS

This work was partly supported by the National Research Foundation of Korea (NRF) grant funded by the Korea government (*MSIT) (No.2018R1A5A7059549) and the Institute of Information & communications Technology Planning & Evaluation (IITP) grant funded by the Korea government (MSIT) (No.2020-0-01373, Artificial Intelligence Graduate School Program (Hanyang University) ). *Ministry of Science and ICT. This work was supported by the National Natural Science Foundation of China under Grant 62376080, the Zhejiang Provincial Major Research and Development Project of China under Grant 2023C01242, the Central Guiding Local Science and Technology Development Fund Projects of China under Grant 2023ZY1008, and the Zhejiang Provincial Key Lab of Equipment Electronics.

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

## APPENDIX

## A    MORE IMPLEMENTATION DETAILS

**Model Architecture.** The presented category-unified models SiamCUT and MoCUT are built upon P2B (Qi et al., 2020) and M$^2$Track (Zheng et al., 2022a) frameworks, respectively. We employ the proposed AdaFormer the feature extraction network for both models. In addition, unified model inputs and learning objectives are integrated into the models, introducing no additional computational overhead during the inference phase. The remaining components of our category-unified models adhere to the structures utilized in P2B and M$^2$Track.

**Training & Inference.** Our category-unified models are trained in a end-to-end manner, using a Tesla A100 GPU. Following common practice (Qi et al., 2020; Zheng et al., 2022a; Hui et al., 2022; Xu et al., 2023), we conduct separate training and testing on the KITTI (Geiger et al., 2012) and NuScenes (Caesar et al., 2020) datasets. To ensure a fair comparison, we train our models on the training sets of all categories within a specific dataset and then tested on the corresponding test sets. We further perform an evaluation of the model pre-trained from KITTI on the Waymo Open Dataset (Sun et al., 2020). The training and inference hyper-parameters for SiamCUT and MoCUT remain consistent with P2B and M$^2$Track, such as optimizer, learning rate, and batch size.

## B    MORE ANALYSIS AND EXPERIMENTS

Here, we first introduce the motivation of our category-unified models in Section B.1. Then we provide the detailed experiments and analysis of the proposed components within our category-unified models, including unified representation network AdaFormer, model inputs and learning objective in Section B.2. These experiments are conducted using Siamese paradigm (Qi et al., 2020) on the KITTI (Geiger et al., 2012) dataset. Finally, we present state-of-the-art comparison on Waymo Open Dataset (Sun et al., 2020) and computational cost analysis in Section B.3 and B.4.

### B.1    MOTIVATION ANALYSIS

Our goal is to develop a category-agnostic model capable of tracking any object regardless of its category. However, different object categories possess distinct attributes, such as shape, size, and motion state, which pose challenges to achieving category unification. As shown in Fig. 5, we visualize the length, width, and relative motion ($\Delta x$, $\Delta y$, $\Delta z$, $\Delta \theta$) of two adjacent frames of all data on Car and Pedestrian category from KITTI (Geiger et al., 2012) dataset. It can be observed that cars exhibit much larger sizes than pedestrians, with different length-to-width ratios. Moreover, the motion states of cars and pedestrians display significant differences, particularly in the $xy$-plane displacement. In addition, we visualize the distribution of background distractors around pedestrians and vehicles, as shown in Fig. 6. The density of distractors surrounding pedestrians is considerably higher than that around vehicles. Considering these factors, we think that handling objects of different categories with diverse sizes, shapes, and motion states uniformly is pivotal for realizing category-unified tracking.

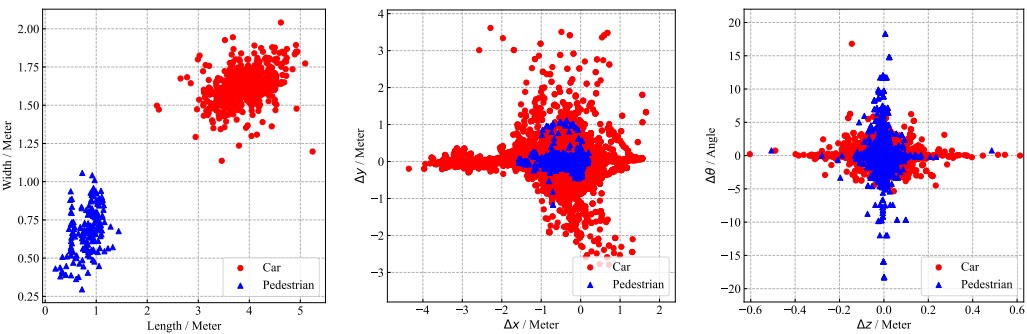

Figure 5: Numerical statistics of object's length, width and relative motion ($\Delta x$, $\Delta y$, $\Delta z$, $\Delta \theta$) of two adjacent frames on Car and Pedestrian categories from KITTI dataset.

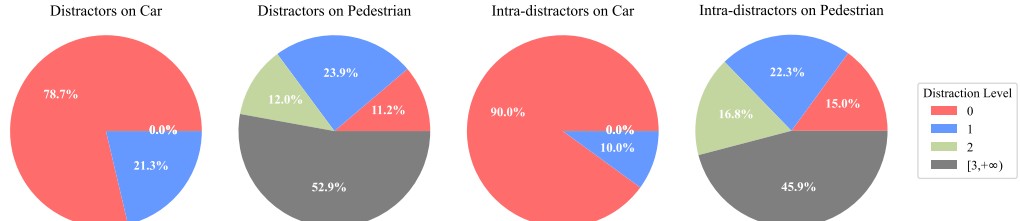

Figure 6: Numerical statistics of distractors and intra-class distractors on Car and Pedestrian categories from KITTI dataset. We plot four quantity levels: 0, 1, 2, and ≥3, with respect to the proportion of the corresponding data size.

### B.2 DETAILED EXPERIMENT AND ANALYSIS OF UNIFIED COMPONENTS

**Unified Representation Network: AdaFormer.** Our unified representation network *AdaFormer* incorporates a group regression module for learning deformable groups to enable adaptive receptive fields for various object categories, along with a vector-attention mechanism to facilitate feature interaction of points within these deformable groups, ultimately forming a category-unified feature representation. Tab. 4 presents an ablation study to understand the two sub-components. Benefiting from the adaptive receptive fields achieved by the group regression module, our representation network can learn geometric information of various object categories in a unified manner. Consequently, when this module is removed, average performance drops by 6.9% and 7.0% in terms of *Success* and *Precision*, respectively. It's noteworthy that the most obvious performance degradation occurs in the Pedestrian category. This is due to the relatively small training samples for the Pedestrian category and the significant differences in shape and size compared to other object categories. To visually understand of how the group regression module works, we provide some visualizations of deformable groups on the Car and Pedestrian categories, as shown in Fig. 7. In addition, when removing the vector-attention mechanism, we employ a feature propagation operator in existing backbone network (Qi et al., 2017a;b) to substitute it. Tab. 4 demonstrates that the vector-attention mechanism plays a crucial role in promoting the learning of a unified feature representation.

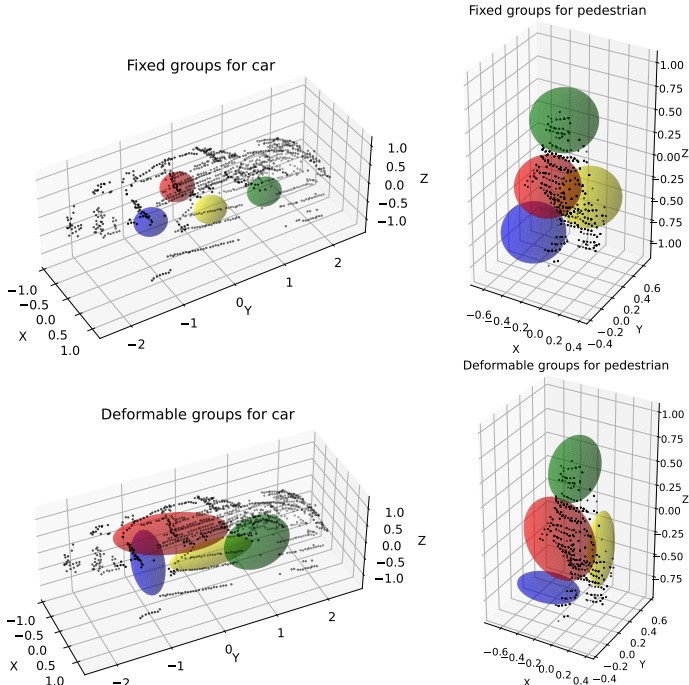

Figure 7: Comparison of groups on the Car and Pedestrian categories. We plot four fixed groups and deformable groups from the first layer in PointNet++ (Qi et al., 2017b) and our *AdaFormer* network, respectively, using different colors.

Table 4: Ablation study of unified representation network. *Success / Precision* are used for evaluation. **Bold** denote the best performance.

| Group Regression Module | Vector-Attention Mechanism | Car [6,424] | Pedestrian [6,088] | Van [1,248] | Cyclist [308] | Mean [14,068] |
|---|---|---|---|---|---|---|
| ✗ | ✗ | 56.3 / 72.4 | 33.2 / 60.4 | 57.0 / 68.6 | 32.2 / 43.5 | 45.9 / 63.2 |
| ✗ | ✓ | 56.5 / 72.7 | 35.2 / 62.9 | 59.4 / 69.3 | 32.3 / 44.0 | 47.1 / 67.6 |
| ✓ | ✗ | 57.7 / 73.8 | 44.9 / 71.2 | 61.8 / 72.0 | 35.3 / 46.1 | 52.1 / 72.0 |
| ✓ | ✓ | **58.1 / 73.9** | **48.2 / 76.2** | **63.1 / 74.9** | **36.7 / 47.4** | **54.0 / 74.6** |

**Unified Model Input.** The scale factor $\alpha$ is an important hyper-parameter in our unified model inputs. Hence, we conduct an ablation experiment using different values to determine the optimal setting for this parameter. As presented in Tab. 5, our method is not sensitive to the scale factor within a reasonable range of values, *i.e.*, when this parameter is set in the range from 0.8 to 1.4. Nevertheless, excessively large value will introduce noise, whereas overly small value will ignore valuable information, both leading to significant performance degradation.

Table 5: Performance using different scale factor $\alpha$. *Success / Precision* are used for evaluation. **Bold** denote the best performance.

| Scale Factor $\alpha$ | Car [6,424] | Pedestrian [6,088] | Van [1,248] | Cyclist [308] | Mean [14,068] |
|---|---|---|---|---|---|
| 0.6 | 52.4 / 68.1 | 42.3 / 70.3 | 48.9 / 57.4 | 31.2 / 43.0 | 47.3 / 67.6 |
| 0.8 | 57.5 / 73.2 | **48.4 / 76.7** | 63.0 / 74.7 | **37.2 / 45.0** | 53.7 / 74.3 |
| 1.0 | 58.1 / **73.9** | 48.2 / 76.2 | **63.1 / 74.9** | 36.7 / **47.4** | **54.0 / 74.6** |
| 1.2 | **58.3** / 73.7 | 48.0 / 76.1 | 62.8 / 74.7 | 36.1 / 46.6 | 53.8 / 74.2 |
| 1.4 | 57.8 / 73.1 | 47.6 / 75.3 | 62.0 / 73.8 | 35.0 / 44.8 | 53.3 / 73.6 |
| 1.6 | 55.8 / 71.5 | 32.4 / 57.0 | 56.4 / 66.2 | 30.2 / 42.5 | 45.2 / 64.2 |

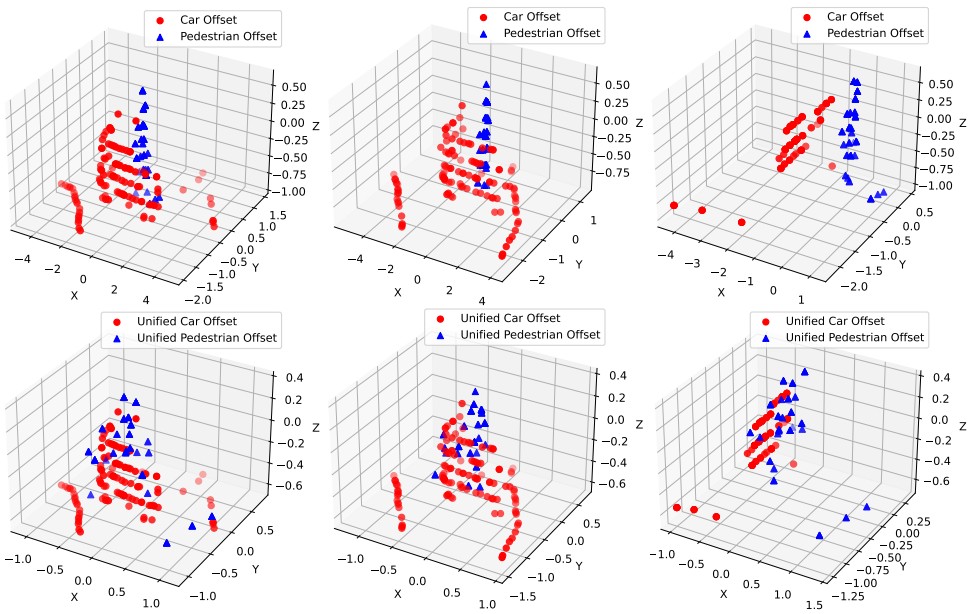

Figure 8: Comparison of offsets between Car and Pedestrian categories. The upper and lower rows represent the cases of without and with unified prediction target design, respectively.

**Unified Learning Objective.** The unified learning objective involves a consistent numerical distribution of predicted targets and a balanced distribution of positive and negative samples. To investigate their contributions, we conduct ablation experiments and report the ablation results in Tab. 6. Firstly, as illustrated in the upper row of Fig. 8, different object categories exhibit significant variations in offset targets, distracting the model. However, by unifying the offset targets across the three coordinate axes $xyz$ based on length, width, and height information (as shown in the lower row), the offset targets of diverse object categories converge within a common numerical space, thereby resulting in improved performance.

In addition, we employ shape-aware labels to define positive and negative samples, which further enhances the tracking performance by 1.2% and 1.4% in average *Success* and *Precision*, as shown in Tab. 6. The scale factor $\beta$ controls the uniform ratio of positive and negative samples. When the parameter value is set too small, it leads to a scarcity of positive samples, especially at the beginning of training, making it difficult for the model to converge. Conversely, setting this value too large results in an overabundance of positive samples, posing a challenge for the model to distinguish the most accurate ones. Therefore, we further conduct an ablation experiment to determine the optimal value for this parameter. According to Tab. 7, we set the scale factor $\beta$ to 0.4 in our main experiment.

Table 6: Ablation study of unified learning objective. *Success* / *Precision* are used for evaluation. **Bold** denote the best performance.

| Unified Prediction Target | Unified Positive -Negative Sample | Car [6,424] | Pedestrian [6,088] | Van [1,248] | Cyclist [308] | Mean [14,068] |
|---|---|---|---|---|---|---|
| ✗ | ✗ | 57.5 / 72.8 | 44.5 / 72.1 | 61.2 / 70.4 | 35.6 / 44.5 | 51.8 / 71.7 |
| ✗ | ✓ | 57.6 / 72.8 | 45.1 / 72.6 | 61.4 / 70.8 | 35.8 / 44.9 | 52.1 / 72.0 |
| ✓ | ✗ | 58.1 / 73.7 | 46.0 / 73.5 | 62.8 / 74.4 | 35.9 / 46.6 | 52.8 / 73.2 |
| ✓ | ✓ | **58.1 / 73.9** | **48.2 / 76.2** | **63.1 / 74.9** | **36.7 / 47.4** | **54.0 / 74.6** |

Table 7: Performance using different scale factor $\beta$. *Success* / *Precision* are used for evaluation. **Bold** denote the best performance.

| Scale Factor $\beta$ | Car [6,424] | Pedestrian [6,088] | Van [1,248] | Cyclist [308] | Mean [14,068] |
|---|---|---|---|---|---|
| 0.2 | 51.4 / 65.8 | 26.6 / 46.9 | 38.4 / 45.0 | 29.6 / 41.8 | 39.1 / 55.3 |
| 0.3 | 55.2 / 71.1 | 34.1 / 61.3 | 48.2 / 56.7 | 31.8 / 44.3 | 45.0 / 65.1 |
| 0.4 | 58.1 / 73.9 | **48.2 / 76.2** | 63.1 / 74.9 | 36.7 / **47.4** | **54.0 / 74.6** |
| 0.5 | **58.3 / 74.1** | 47.7 / 75.8 | **65.7 / 76.0** | **37.2** / 47.3 | 54.0 / 74.5 |
| 0.6 | 56.4 / 72.6 | 44.2 / 73.1 | 58.8 / 67.7 | 34.7 / 44.3 | 50.2 / 71.8 |
| 0.7 | 52.6 / 67.0 | 39.1 / 66.5 | 56.3 / 65.8 | 32.5 / 42.1 | 46.7 / 66.2 |

## B.3 RESULTS ON WAYMO OPEN DATASET

To validate the generalization ability of our proposed category-unified models, we conduct an evaluation by applying the models trained on KITTI dataset to the Waymo Open Dataset (WOD) (Sun et al., 2020). We select six state-of-the-art methods that have reported performance on WOD for comparison. Different from existing methods like CXTrack (Xu et al., 2023) and M$^2$Track (Zheng et al., 2022a) require pre-trained models on both Car and Pedestrian categories from KITTI for tracking vehicle and pedestrian on WOD, our SiamCUT and MoCUT achieve category unification. Therefore, a single model is capable of simultaneously tracking both vehicles and pedestrians. As reported in Tab. 8, our MoCUT consistently outperforms other methods across all subsets, including easy, medium, and hard subsets, for both Vehicle and Pedestrian categories. Compared to baseline category-specific methods P2B and M$^2$Track, our category-unified models exhibit competitive performance, achieving improvements in various subsets. These results demonstrate that our proposed category-unified models SiamCUT and MoCUT not only generalize well across diverse categories but also exhibit a certain level of generalization to datasets with varied data distributions.

Table 8: Performance comparisons with state-of-the-art methods on Waymo Open Dataset. **Bold** and underline represent our category-unified methods and the corresponding baselines.

| Method | Vehicle | | | | Pedestrian | | | |
|---|---|---|---|---|---|---|---|---|
| | Easy [67,832] | Medium [61,252] | Hard [56,647] | Mean [185,731] | Easy [85,280] | Medium [82,253] | Hard [74,219] | Mean [241,752] |
| P2B (Qi et al., 2020) | 57.1 / 65.4 | 52.0 / 60.7 | 47.9 / 58.5 | 52.6 / 61.7 | 18.1 / 30.8 | 17.8 / 30.0 | 17.7 / 29.3 | 17.9 / 30.1 |
| BAT (Zheng et al., 2021) | 61.0 / 68.3 | 53.3 / 60.9 | 48.9 / 57.8 | 54.7 / 62.7 | 19.3 / 32.6 | 17.8 / 29.8 | 17.2 / 28.3 | 18.2 / 30.3 |
| V2B (Hui et al., 2021) | 64.5 / 71.5 | 55.1 / 63.2 | 52.0 / 62.0 | 57.6 / 65.9 | 27.9 / 43.9 | 22.5 / 36.2 | 20.1 / 33.1 | 23.7 / 37.9 |
| STNet (Hui et al., 2022) | 65.9 / 72.7 | 57.5 / 66.0 | 54.6 /64.7 | 59.7 / 68.0 | 29.2 / 45.3 | 24.7 / 38.2 | 22.2 / 35.8 | 25.5 / 39.9 |
| CXTrack (Xu et al., 2023) | 63.9 / 71.1 | 54.2 / 62.7 | 52.1 / 63.7 | 57.1 / 66.1 | 35.4 / 55.3 | 29.7 / 47.9 | 26.3 / 44.4 | 30.7 / 49.4 |
| M$^2$Track (Zheng et al., 2022a) | 68.1 / 75.3 | 58.6 / 66.6 | 55.4 / 64.9 | 61.1 / 69.3 | 35.5 / 54.2 | 30.7 / 48.4 | 29.3 / 45.9 | 32.0 / 49.7 |
| SiamCUT (Ours) | **58.3 / 66.0** | **50.8 / 60.8** | **49.2 / 59.1** | **53.0 / 62.2** | **23.4 / 36.6** | **20.7 / 32.0** | **21.4 / 31.5** | **21.9 / 33.5** |
| MoCUT (Ours) | **68.3 / 75.0** | **59.4 / 66.9** | **57.1 / 66.3** | **61.9 / 69.7** | **36.5 / 54.8** | **30.8 / 48.9** | **29.5 / 45.4** | **32.4 / 49.9** |

## B.4 COMPUTATIONAL COST ANALYSIS

In practical applications, computational cost analysis has always been a critical metric to evaluate the performance of trackers. As shown in Tab. 9, we analyze the model complexity, computational

overhead and inference speed by counting parameters, floating-point-operations per second (FLOPs) and latency/speed, respectively. The parameters and FLOPs for all methods are manually counted, if feasible. The latency and speed for baseline methods P2B (Qi et al., 2020) and M$^2$Track (Zheng et al., 2022a) are computed in our workstation to ensure a fair comparison. The other methods utilize reported results from relative references (Giancola et al., 2019; Zheng et al., 2021; Zhou et al., 2022; Nie et al., 2023b;c; Wang et al., 2023; Xu et al., 2023). Compared to the baseline methods, our category-unified models demonstrate performance improvement with a tolerable computational overhead.

Table 9: Computation cost comparisons with state-of-the-art methods. **Bold** and underline represent our category-unified methods and the corresponding baselines.

| Method | Parameters | FLOPs | Latency | Speed | Hardware | KITTI Performance |
|---|---|---|---|---|---|---|
| SC3D (Giancola et al., 2019) | 6.46 M | 19.82 G | - | 2 Fps | GTX 1080Ti | 32.1 / 48.5 |
| P2B (Qi et al., 2020) | 1.34 M | 4.30 G | 20.8 ms | 48 Fps | RTX 3070Ti | 42.4 / 60.0 |
| BAT (Zheng et al., 2021) | 1.48 M | 2.77 G | - | 57 Fps | RTX 2080 | 51.2 / 72.8 |
| PTTR (Shan et al., 2021) | 2.27 M | 2.61 G | - | 50 Fps | Tesla V100 | 57.9 / 78.2 |
| OSP2B (Nie et al., 2023b) | 1.70 M | 2.57 G | 29.5 ms | 34 Fps | GTX 1080Ti | 60.5 / 82.3 |
| GLT-T (Nie et al., 2023c) | 2.60 M | 3.87 G | 33.4 ms | 30 Fps | GTX 1080Ti | 60.1 / 79.3 |
| PCET (Wang et al., 2023) | - | - | 30.5 ms | 33 Fps | GTX 1080Ti | 64.8 / 81.3 |
| CXTrack (Xu et al., 2023) | 18.3 M | 4.63 G | 29.2 ms | 34 Fps | RTX 3090 | 67.5 / 85.3 |
| M2Track (Zheng et al., 2022a) | 2.24 M | 2.54 G | 15.9 ms | 63 Fps | RTX 3070Ti | 62.9 / 83.4 |
| SiamCUT (Ours) | **2.06 M** | **4.51 G** | **27.5 ms** | **36 Fps** | **RTX 3070Ti** | **54.0 / 74.6** |
| MoCUT (Ours) | **2.34 M** | **3.27 G** | **20.9 ms** | **48 Fps** | **RTX 3070Ti** | **65.8 / 85.0** |

## C    COMPARISON WITH CATEGORY-SPECIFIC MODELS

To further demonstrate the potential of our category-unified models, we integrate the proposed unified components, including unified representation network *AdaFormer*, model inputs and learning objective into existing tracking methods. We select some classic trackers, such as PTT (Shan et al., 2021), PTTR (Zhou et al., 2022) and OSP2B (Nie et al., 2023b) to report the results, as presented in Tab. 10. These unified components not only empower category-specific trackers to track objects across all categories, but also enhance overall tracking performance, which proves the effectiveness and promise of our proposed components.

Table 10: Performance comparisons on the KITTI dataset. *"Improvement"* refers to the performance gain of our category-unified models over the corresponding category-specific counterparts. "▓" and "▓" refer to Siamese and motion-centric paradigms, respectively.

| Method | Car [6,424] | Pedestrian [6,088] | Van [1,248] | Cyclist [308] | Mean [14,068] |
|---|---|---|---|---|---|
| Category-specific P2B (Qi et al., 2020) | 56.2 / 72.8 | 28.7 / 49.6 | 40.8 / 48.4 | 32.1 / 44.7 | 42.4 / 60.0 |
| Category-unified P2B (Ours) | 58.1 / 73.9 | 48.2 / 76.2 | 63.1 / 74.9 | 36.7 / 47.4 | 54.0 / 74.6 |
| Improvement | ↑ 1.9 / ↑ 1.1 | ↑ 19.5 / ↑ 26.6 | ↑ 22.3 / ↑ 26.5 | ↑ 4.6 / ↑ 3.3 | ↑ 11.6 / ↑ 14.6 |
| Category-specific PTT (Shan et al., 2021) | 67.8 / 81.8 | 44.9 / 72.0 | 43.6 / 52.5 | 37.2 / 47.3 | 55.1 / 74.2 |
| Category-unified PTT (Ours) | 67.6 / 82.1 | 49.2 / 77.4 | 65.4 / 77.0 | 37.5 / 46.8 | 58.8 / 76.4 |
| Improvement | ↓ 0.2 / ↑ 0.3 | ↑ 4.3 / ↑ 4.6 | ↑ 21.8 / ↑ 24.5 | ↑ 0.3 / ↓ 0.5 | ↑ 3.7 / ↑ 2.2 |
| Category-specific PTTR (Zhou et al., 2022) | 65.2 / 77.4 | 50.9 / 81.6 | 52.5 / 61.8 | 65.1 / 90.5 | 57.9 / 78.2 |
| Category-unified PTTR (Ours) | 68.3 / 80.1 | 53.7 / 84.1 | 64.2 / 75.6 | 66.8 / 93.2 | 61.6 / 81.8 |
| Improvement | ↑ 3.1 / ↑ 2.7 | ↑ 2.8 / ↑ 2.5 | ↑ 11.7 / ↑ 13.8 | ↑ 1.7 / ↑ 2.7 | ↑ 3.7 / ↑ 3.6 |
| Category-specific OSP2B (Nie et al., 2023b) | 67.5 / 82.3 | 53.6 / 85.1 | 56.3 / 66.2 | 65.6 / 90.5 | 60.5 / 82.3 |
| Category-unified OSP2B (Ours) | 67.5 / 82.8 | 55.1 / 86.7 | 68.7 / 79.3 | 65.4 / 91.2 | 62.3 / 84.4 |
| Improvement | ↑ 1.0 / ↑ 0.5 | ↑ 1.5 / ↑ 1.6 | ↑ 12.4 / ↑ 13.1 | ↓ 0.2 / ↑ 0.7 | ↑ 1.8 / ↑ 2.1 |
| Category-specific M$^2$Track (Zheng et al., 2022a) | 65.5 / 80.8 | 61.5 / 88.2 | 53.8 / 70.7 | 73.2 / 93.5 | 62.9 / 83.4 |
| Category-unified M$^2$Track (Ours) | 67.6 / 80.5 | 63.3 / 90.0 | 64.5 / 78.8 | 76.7 / 94.2 | 65.8 / 85.0 |
| Improvement | ↑ 1.1 / ↓ 0.3 | ↑ 1.8 / ↑ 1.8 | ↑ 9.7 / ↑ 8.1 | ↑ 3.5 / ↑ 1.3 | ↑ 2.9 / ↑ 1.6 |

