# OpenReview forum: "Towards Category Unification of 3D Single Object Tracking on Point Clouds"
_ICLR.cc/2024/Conference — ICLR 2024 poster_

### Official Review · Reviewer_zjK7 · 2023-10-22

**Soundness:** 3 good
**Presentation:** 3 good
**Contribution:** 3 good
**Rating:** 6
**Confidence:** 4

**Summary:**

This paper introduces an Adaformer to dynamically extract features of different categories with diverse object sizes. To this end, it can adapt to LiDAR-based 3D SOT tasks for unified training. Albeit the idea is relatively common, it can achieve the target of unified training with performance improvement.

**Strengths:**

- It is the first work to unified train all categories on the 3D SOT task.
- The proposed Adaformer can effectively learn adaptively ball region relative to different categories, which agrees with the motivation of this paper.
- The paper is well-written.

**Weaknesses:**

- You should discuss relative works on your key idea (e.g., encoding shape- and size-changed geometric information).
   Relative works include [a] (adaptive region learning), [b] (dynamic ball-query selection for size, dynamic foreground and background learning), [c], etc.

   [a] Pyramid r-cnn: Towards better performance and adaptability for 3d object detection.

   [b] DBQ-SSD: Dynamic Ball Query for Efficient 3D Object Detection.

   [c] RBGNet: Ray-Based Grouping for 3D Object Detection.

- The comparison of latency should include other methods that have been reported, e.g., [c], [d], etc. In addition, the recent SOTA SOT methods should also be included.

   [c] Beyond 3d siamese tracking: A motion-centric paradigm for 3d single object tracking in point clouds.

   [d] Implicit and Efficient Point Cloud Completion for 3D Single Object Tracking.

- Why not verify on Waymo dataset？

- Can Adaformer be extended to general detection？

- Can you conduct other statistical analysis to reveal diverse size of different categories to further support you motivation?

**Questions:**

Please see the Weaknesses.

---

> ### Author Response · Authors · 2023-11-20
> **Related works about idea, latency comparison, recent methods comparison, verification on Waymo, AdaFormer for general detection and statistical analysis for supporting motivation**
>
> Thanks for your valuable comment very much. We have polished our paper according to your suggestions. Here, we number and address questions as follows. Changes are highlighted in $\textcolor{red}{red}$ in the revised manuscript.
>
> (1) Related works about idea.
> According to your insightful comment, we have discussed related works on our key idea to give a better understanding of our work. Our core idea is to encode adaptive group configurations to accommodate shape- and size-changed geometric information from cross-category data. We appreciate your observation regarding the connection between the mentioned references and our core idea, specifically the dynamic receptive field. We have added a subsection, as shown in the $\textbf{“2.3 Dynamic Grouping” subsection of the main manuscript on page 3, line 136}$.
>
> (2) Latency comparison.
> According to your comment, we have conducted a comparison of latency between our unified models and eight state-of-the-art methods (including the methods you mentioned), aiming to give a comprehensive assessment of the proposed methods in comparison to existing methods. In addition, we have also included a comparison of model parameters and FLOPs. Our methods introduce slight overhead compared to the baseline methods. For a detailed comparison and analysis, please refer to the $\textbf{“B.4 Computational Cost Analysis” subsection of the Appendix on page 16, line 585}$.
>
> (3) Recent methods comparison.
> To better demonstrate the effectiveness of the proposed category-unified models, we have compared the recent SOTA methods, such as TAT [ACCV2022], STNet [ECCV2022], DMT [T-ITS2023], CXTrack [CVPR2023] and SyncTrack [ICCV2023] for comparison, as shown in the $\textbf{“4.2 Comparison with State-of-the-art Methods” subsection of the main manuscript on page 7, line 295}$.
>
> (4) Verification on Waymo.
> In the community, KITTI and NuScenes are often employed for a comprehensive evaluation of a tracker. Specifically, tracking models are typically trained and tested on each category within these two datasets. In contrast, Waymo is often utilized to assess the generalization capabilities of the models. A common practice is to test a model pretrained from KITTI on the Waymo dataset. To further validate the generalization of our unified model and enhance the quality of our work, we have evaluated the proposed category-unified model on Waymo, following common experiment setup. Detailed experiments and analyses are provided in the $\textbf{“B.3 Results on Waymo Open Dataset” subsection of the Appendix on page 16, line 571}$.
>
> (5) AdaFormer for general detection.
> Thanks for your meaningful comment. We believe that the concept of AdaFormer holds potential for extension to general detection. To the best of our knowledge, 3D point cloud-based object detection can be broadly categorized into indoor and outdoor (autonomous driving scenarios) detection. From a model perspective, detection methods can be further classified into those relying on point representation and those utilizing voxel representation. Currently, in indoor scenes, methods based on point representation dominate, while in outdoor scenes, voxel representation-based methods prevail due to the computational challenges posed by large-scale scenarios for point representation-based methods. Our AdaFormer is a point representation-based feature extraction network. Its design philosophy may be extended to point representation-based detection methods, introducing dynamic receptive fields to enhance detection accuracy. For detection methods based on voxel representation, our approach may also offer insights, such as dynamically adjusting voxel sizes to enhance feature representation for objects with diverse shapes and sizes.
>
> (6) Statistical analysis for supporting motivation.
> We can provide other statistical analysis to reveal diverse size of different categories to further support our motivation. Our goal is to design a category-unified model capable of tracking any object regardless of its category. The challenge we face is that different object categories exhibit distinct data distributions (size and shape) and learning objectives. Therefore, our motivation is to handle different object categories with diverse sizes and shapes in a unified manner for category-unified tracking. In the Appendix of the old manuscript, we have already visualized the differences in learning objectives as shown in Fig. 8 (in the new manuscript). To support our motivation from various perspectives, we have visualized the statistical analysis of size distributions, motion state distributions and background distraction distributions for the Car and Pedestrian categories on KITTI, and then performed a comparison analysis between the two categories. The presented statistical analysis can effectively support our motivation. Details can be found in the $\textbf{“B.1 Motivation Analysis” subsection of the Appendix on page 13, line 519}$.

---

### Official Review · Reviewer_vKoh · 2023-10-31

**Soundness:** 3 good
**Presentation:** 4 excellent
**Contribution:** 3 good
**Rating:** 8
**Confidence:** 4

**Summary:**

Previous 3D single object tracking models are all category-specific, incurring redundant parameters. In this paper, the authors propose to unify the different categories in a single model. A novel point cloud representation learning network based on transformers, named AdaFormer, is proposed to encode the dynamically varying shape and size information from cross-category data in a unified manner. Moreover, the authors construct two unified models following previous Siamese and motion-centric paradigms to compare. Performance gains validate the effectiveness of proposed method.

**Strengths:**

1. The motivation is great, category-unified model designing is meaningful and significant to SOT task and trying to unify them is of novelty.
2. The model achieves good performance on KITTI and nuScenes.

**Weaknesses:**

No obvious weakness from my perspective

**Questions:**

Are there any inference speed and FLOPs comparison for proposed method?

---

> ### Author Response · Authors · 2023-11-20
> **Inference speed and FLOPs comparison**
>
> Thanks very much for your positive feedback, we have polished our paper according to your valuable suggestions. We have incorporated a comparison of inference speed and FLOPs to enhance the quality of our work. We have also presented the model parameters. The proposed category-unified models introduce a tolerable additional inference cost and computational overhead compared to baseline methods (For example, P2B [48 Fps, 4.30 GFLOPs, 1.34M parameters] v.s. SiamCUT [36 Fps, 4.51 GFLOPs, 2.06M parameters]), mainly incurred by the proposed AdaFormer backbone. Our unified model inputs and learning objective do not incur any additional overhead during the testing phase. Detailed comparisons and analyses are provided in the $\textbf{“B.4 Computational Cost Analysis” subsection of the Appendix on page 16, line 585, which is highlighted in \textcolor{red}{red} in the revised manuscript}$.

---

> > ### Comment · Reviewer_vKoh · 2023-11-22
> > **Response to authors**
> >
> > Thank you for your response. I think the computational cost is tolerable, and expect further work to accelerate inference speed. I will keep my score.

---

> > > ### Author Response · Authors · 2023-11-22
> > > **Response to reviewer**
> > >
> > > Thank you for your reply and affirmation, we will stay tuned to accelerate the inference of category-unified tracking model.

---

### Official Review · Reviewer_5oVT · 2023-11-01

**Soundness:** 3 good
**Presentation:** 3 good
**Contribution:** 3 good
**Rating:** 6
**Confidence:** 5

**Summary:**

Existing 3D single object tracking (SOT) approaches mainly focus on category-specific model training and evaluation. Inspired by general 2D SOT, this paper proposes to use category-unified model for 3D SOT. Specifically, the paper proposes a point set network named AdaFormer to encode geometric information of different object categories in a unified manner. To further boost the performance, the unified model inputs and learning objective are introduced to facilitate the learning of unified representation. To verify the effectiveness, two category-unified models SiamCUT and MoCUT based on Siamese and motion-centric 3D SOT paradigms are proposed. Experiments on KITTI and NuScenes datasets show that the proposed approaches gain much performance improvements over the baselines.

**Strengths:**

- The paper written&&organization is good, which is easy to follow.
- I think the problem solved in this paper is valuable to the 3D SOT community, since current approaches mainly need to train multiple models corresponding to various training categories in the dataset in order to achieve higher performance. This paper shows competitive category-specific results by only learning a unified representation model (although I do not see the authors claim any training details about their unified training, e.g., training their models on all category samples on KITTI and then test it on category-specific KITTI).
- The paper is technically sound, which solves the above problem progressively by proposing multiple modules.

**Weaknesses:**

- The proposed approach can track objects across all categories using a single network with shared parameters. But as I mentioned above, there is no training details about the unified training. Is the proposed only trained on the full KITTI and then test on it (the same for Nuscenes)? or the combination of KITTI and Nuscenes are used? Please all more illustration in the paper.
- The main concern in this paper is about the lack of Waymo dataset evaluation and missing recent approaches for comparison. Please also include the latest references below for comparison, in order to better verify the effectiveness of the proposed approach.

[1] Temporal-aware Siamese Tracker: Integrate Temporal Context for 3D Object Tracking. ACCV 2022.
[2] 3D Siamese Transformer Network for Single Object Tracking on Point Clouds. ECCV 2022.
[3] CXTrack: Improving 3D Point Cloud Tracking with Contextual Information. CVPR 2023.
[4] A Lightweight and Detector-Free 3D Single Object Tracker on Point Clouds. IEEE Transactions on Intelligent Transportation Systems. 2023.

**Questions:**

- The proposed approach can track objects across all categories using a single network with shared parameters. But as I mentioned above, there is no training details about the unified training. Is the proposed only trained on the full KITTI and then test on it (the same for Nuscenes)? or the combination of KITTI and Nuscenes are used? Please all more illustration in the paper.
- The main concern in this paper is about the lack of Waymo dataset evaluation and missing recent approaches for comparison. Please also include the latest references below for comparison, in order to better verify the effectiveness of the proposed approach.

[1] Temporal-aware Siamese Tracker: Integrate Temporal Context for 3D Object Tracking. ACCV 2022.
[2] 3D Siamese Transformer Network for Single Object Tracking on Point Clouds. ECCV 2022.
[3] CXTrack: Improving 3D Point Cloud Tracking with Contextual Information. CVPR 2023.
[4] A Lightweight and Detector-Free 3D Single Object Tracker on Point Clouds. IEEE Transactions on Intelligent Transportation Systems. 2023.

---

> ### Author Response · Authors · 2023-11-20
> **Details of unified training, evaluation on Waymo and recent methods comparison**
>
> Thanks for your valuable comment very much. We have polished our paper according to your suggestions. Here, we number and address questions as follows. Changes are highlighted in $\textcolor{red}{red}$ in the revised manuscript.
>
> (1) Details of unified training.
> In our experimental configuration, the proposed unified model is trained on the full KITTI and then test on it, the same for NuScenes, not the combination of KITTI and NuScenes. More specifically, the experimental setup commonly used in the community is to train and test each category individually within each dataset. To be a fair comparison, our unified model is trained on all training data (cross-category data) of a dataset, and then tested on all test data (diverse categories) of that dataset. In addition, during the training phase, we directly use data from all categories to train a unified model, without any additional data augmentation techniques. We appreciate the insightful comment and have made improvements to the manuscript by supplementing training details, as shown in the $\textbf{“A More Implementation Details” section of the Appendix on page 13, line 497}$.
>
> Further discussion: There are notable differences among various datasets. For example, NuScenes displays more pronounced sparsity compared to KITTI and Waymo. We posit that cross-dataset training requires addressing challenges related to data distribution. This may include dealing with diverse background and motion state distributions, such as differences in pedestrian behavior between KITTI and NuScenes. The concept of joint cross-dataset and cross-category training will enable a single model to track all objects across all categories, irrespective of data distribution, which further enhances the model's generalization capabilities. In the future, we will keep it tune and further improve our unified model.
>
> (2) Evaluation on Waymo.
> According to your comment, we have evaluated the proposed method on Waymo. Unlike KITTI and NuScenes, the community often conducts generalization experiments on Waymo dataset. Following this, we test the KITTI pre-trained model on Waymo. We also compare our category-unified models with existing SOTA methods, including the recent methods you mentioned. The experimental results demonstrate that the proposed category-unified model not only generalizes well across diverse categories but also exhibits a certain level of generalization to datasets with varied data distributions. Detailed experiments and analyses are provided in the $\textbf{“B.3 Results on Waymo Open Dataset” subsection of the Appendix on page 16, line 571}$.
>
> (3) Recent methods comparison.
> To better verify the effectiveness of the proposed approach, we have included all the recent methods you mentioned and the latest SyncTrack [ICCV2023] for comparison, and added the corresponding comparison analysis, as shown in the $\textbf{“4.2 Comparison with State-of-the-art Methods” subsection of the main manuscript on page 7, line 295}$. In addition, we also compare our category-unified models with recent SOTA methods on Waymo, as shown in the $\textbf{“B.3 Results on Waymo Open Dataset” subsection of the Appendix on page 16, line 571}$.

---

> > ### Comment · Reviewer_5oVT · 2023-11-22
> >
> > Thank you for your detailed feedback. The revised paper addresses my concerns.

---

> > > ### Author Response · Authors · 2023-11-22
> > > **Response to reviewer**
> > >
> > > Thank you for your reply. Looking forward to further discussion with you in the future.

---

### Meta-Review · Area_Chair_TsJf · 2023-12-09

**Metareview:**

The paper was reviewed by 3 experts, with initially positive reviews (6,6,8). The major concerns were:

1. missing training details.
2. comparison with some recent methods missing, and comparisons on Waymo evaluation missing.
3. missing runtime information.
4. missing related works.
5. statistical analysis to support the motivation?
6. applicability to general detection?

The authors wrote a response to address the concerns. Specifically, they provided more details, results on Waymo, runtime information, and additional analysis. Overall, the reviewers were satirized with the response, and appreciated the value of proposing a unified model to the 3D SOT community, as well as the competitive results compared to category-specific models.

The AC agrees with the reviewers, and thus recommends accept.  Authors should prepare a camera-ready version according to the reviews and discussion.

**Justification For Why Not Higher Score:**

- it is not a new problem, but a re-definition of an existing problem.
- the proposed pipeline follows the structure of other 3D SOT pipelines.

**Justification For Why Not Lower Score:**

the move towards unified tracking has value to the community. the results are good.

---

### Decision · Program_Chairs · 2024-01-16

Accept (poster)